# LEARNING TO INVERT: SIMPLE ADAPTIVE ATTACKS FOR GRADIENT INVERSION IN FEDERATED LEARNING

## ABSTRACT

Gradient inversion attack enables recovery of training samples from model updates in federated learning (FL) and constitutes a serious threat to data privacy. To mitigate this vulnerability, prior work proposed both principled defenses based on differential privacy, as well as heuristic defenses based on gradient compression as countermeasures. These defenses have so far been very effective, in particular those based on gradient compression that allow the model to maintain high accuracy while greatly reducing the attack's effectiveness. In this work, we argue that such findings do not accurately reflect the privacy risk in FL, and show that existing defenses can be broken by a simple adaptive attack that trains a model using auxiliary data to learn how to invert gradients on both vision and language tasks.

## 1 INTRODUCTION

Federated learning (FL; (McMahan et al., 2017)) is a popular framework for distributed model training on sensitive user data. Instead of centrally storing the training data, FL operates in a server-client setting where the server hosts the model and has no direct access to the data. The clients can apply the model on their private data and send gradient updates back to the server. This learning regime promises data privacy as users only share gradients but never any raw data. However, recent work (Zhu et al., 2019; Zhao et al., 2020; Geiping et al., 2020) showed that despite these efforts, the server can still recover the training data from gradient updates, violating the promise of data privacy in FL. These so-called *gradient inversion attacks* operate by optimizing over the input space to find training samples whose gradient matches that of the observed gradient, and such attacks remain effective even when clients utilize secure aggregation (Bonawitz et al., 2016) to avoid revealing individual updates (Yin et al., 2021; Jeon et al., 2021).

As countermeasures against these gradient inversion attacks, prior work proposed both principled defenses based on differential privacy (Abadi et al., 2016), as well as heuristics that compress the gradient update through gradient pruning (Aji & Heafield, 2017) or sign compression (Bernstein et al., 2018). In particular, gradient compression defenses have so far enjoyed great success, severely hindering the effectiveness of existing optimization-based attacks (Zhu et al., 2019; Jeon et al., 2021) while maintaining close to the same level of accuracy for the trained model. As a result, these limitations seemingly diminish the threat of gradient inversion in practical FL applications.

In this paper we argue that evaluating defenses on existing optimization-based attacks may provide a false sense of security. To this end, we propose a simple *learning-based* attack—which we call *Learning To Invert* (LTI)—that trains a model to learn how to invert the gradient update to recover client samples; see Figure 1 for an illustration. We assume that the adversary (*i.e.*, the server) has access to an *auxiliary dataset* whose distribution is similar to that of the private data, and use it to generate training samples for the gradient inversion model by querying the global model for gradients. Our attack is highly adaptable to different defenses since applying a defense simply amounts to training data augmentation for the gradient inversion model.

We empirically demonstrate that LTI can successfully circumvent defenses based on gradient perturbation (*i.e.*, using differential privacy; (Abadi et al., 2016)), gradient pruning (Aji & Heafield, 2017) and sign compression (Bernstein et al., 2018) on both vision and language tasks.

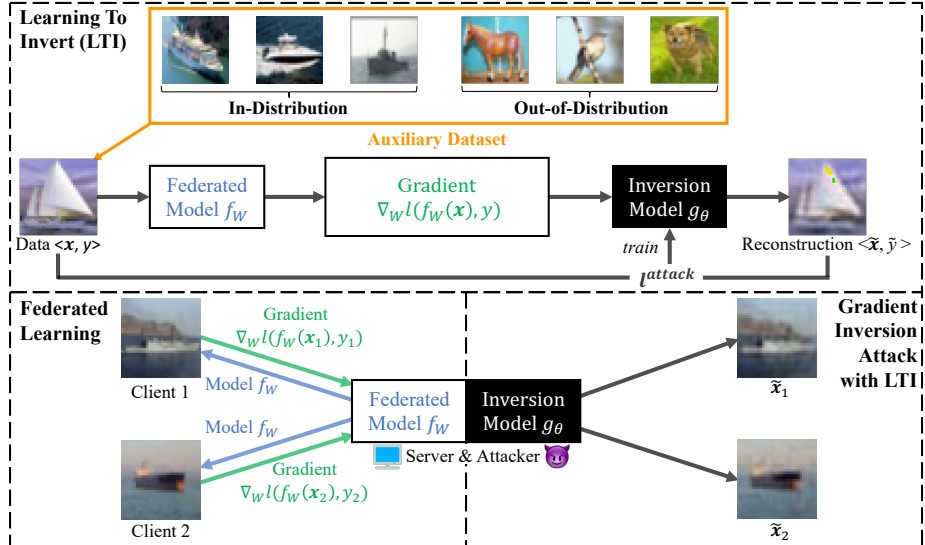

Figure 1: Illustration of federated learning (FL) and gradient inversion methods. The goal of gradient inversion is to recover training data $(\mathbf{x}, y)$ from the observed gradient $\nabla_{\mathbf{w}} \ell(f_{\mathbf{w}}(\mathbf{x}), y)$. Optimization-based methods (*e.g.*, (Zhu et al., 2019; Geiping et al., 2020; Yin et al., 2021; Jeon et al., 2021)) directly optimize $(\tilde{\mathbf{x}}, \tilde{y})$ in search for a sample that produces gradient similar to that of $(\mathbf{x}, y)$. Our proposed learning-based approach, which we call *Learning to Invert*, instead trains an inversion model $g_\theta$ to reconstruct training samples from their gradient.

- Vision: We evaluate on the CIFAR10 (Krizhevsky et al., 2009) classification dataset. LTI attains recovery accuracy close to that of the best optimization-based method when no defense is applied, and significantly outperforms all prior attacks under defense.
- NLP: We experiment with causal language model training on the WikiText (Merity et al., 2016) dataset, where LTI attains state-of-the-art performance in all settings, with or without defense.

Given the strong empirical performance of LTI and its adaptability to different learning tasks and defense mechanisms, we advocate for its use as a simple baseline for future studies on gradient inversion attacks in FL.

## 2   BACKGROUND

**Federated learning.**   The objective of federated learning (McMahan et al., 2017) is to train a machine learning model in a distributed fashion without centralized collection of training data. In detail, let $f_{\mathbf{w}}$ be the *global model* parameterized by $\mathbf{w}$, and consider a supervised learning setting that optimizes $\mathbf{w}$ by minimizing a loss function $\ell$ over the training set $\mathcal{D}_{\text{train}}$: $\sum_{(\mathbf{x},y) \in \mathcal{D}_{\text{train}}} \ell(f_{\mathbf{w}}(\mathbf{x}), y)$. In centralized learning this is typically done by computing a stochastic gradient $\frac{1}{B} \sum_{i=1}^{B} \nabla_{\mathbf{w}} \ell(f_{\mathbf{w}}(\mathbf{x}_i), y_i)$ over a randomly drawn batch of data $(\mathbf{x}_1, y_1), \ldots, (\mathbf{x}_B, y_B)$ and minimizing $\ell$ using gradient descent.

In FL, instead of centrally collecting $\mathcal{D}_{\text{train}}$ to draw a random batch during training, the training set $\mathcal{D}_{\text{train}}$ is distributed across multiple clients and the model $f_{\mathbf{w}}$ is stored on a central server. At each iteration, the model parameter $\mathbf{w}$ is transmitted to each client to compute the per-sample gradients $\{\nabla_{\mathbf{w}} \ell(f_{\mathbf{w}}(\mathbf{x}_i), y_i)\}_{i=1}^{B}$ locally over a set of clients. The server and clients then execute a *federated aggregation* protocol to compute the average gradient for the gradient descent update. A major advantage of FL is data privacy since clients do not need to disclose their data explicitly, but rather only send their gradient $\nabla_{\mathbf{w}} \ell(f_{\mathbf{w}}(\mathbf{x}_i), y_i)$ to the server. Techniques such as secure aggregation (Bonawitz et al., 2016) and differential privacy (Dwork et al., 2006; 2014) can further reduce the privacy leakage from sending this gradient update.

**Gradient inversion attack.** Despite the promise of data privacy in FL, recent work showed that the heuristic of sending gradient updates instead of training samples themselves in fact provides a false sense of security. Zhu et al. (2019) showed in their seminal paper that it is possible for the server to recover the full batch of training samples given aggregated gradients. These *optimization-based* gradient inversion attacks operate by optimizing a set of *dummy data* $\tilde{\mathbf{x}}_1, \ldots, \tilde{\mathbf{x}}_B$ and labels $\tilde{y}_1, \ldots, \tilde{y}_B$ to match their gradient to the observed gradient:

$$\min_{\tilde{\mathbf{x}}} \left\| \sum_{i=1}^{B} \nabla_{\mathbf{w}} \ell(f_{\mathbf{w}}(\tilde{\mathbf{x}}_i), \tilde{y}_i) - \sum_{i=1}^{B} \nabla_{\mathbf{w}} \ell(f_{\mathbf{w}}(\mathbf{x}_i), y_i) \right\|_2^2. \tag{1}$$

For image tasks, since Equation 1 is differentiable in $\tilde{\mathbf{x}}_i$ and $\tilde{y}_i$ and the model parameter $\mathbf{w}$ is known to the server, the server can optimize Equation 1 using gradient-based search. Doing so yields recovered samples $(\tilde{\mathbf{x}}_i, \tilde{y}_i)$ that closely resemble actual samples $(\mathbf{x}_i, y_i)$ in the batch. In practice this approach is highly effective, and follow-up works proposed several optimizations to further improve its recovery accuracy (Geiping et al., 2020; Yin et al., 2021; Jeon et al., 2021).

For language tasks this optimization problem is considerably more complex since the samples $\mathbf{x}_1, \ldots, \mathbf{x}_B$ are sequences of discrete tokens, and optimizing Equation 1 amounts to solving a discrete optimization problem. To circumvent this difficulty, Zhu et al. (2019) and Deng et al. (2021) instead optimize the *token embeddings* to match the observed gradient and then maps the recovered embeddings to their closest tokens in the embedding layer to recover the private text. In contrast, Gupta et al. (2022) leveraged the insight that gradient of the token embedding layer can be used to recover exactly the set of tokens present in the training sample, and then uses beam search to optimize the ordering of tokens for fluency to recover the private text.

**Gradient inversion under the malicious server setting.** The aforementioned gradient inversion attacks operate under the *honest-but-curious* setting where the server faithfully executes the federated learning protocol, but attempts to extract private information from the observed gradients. Fowl et al. (2021), Boenisch et al. (2021) and Fowl et al. (2022) consider a stronger *malicious server* threat model that allows the server to transmit arbitrary model parameters $\mathbf{w}$ to the clients. Under this threat model, it is possible to carefully craft the model parameters so that the training sample can be recovered exactly from its gradient even when the batch size $B$ is large. While this setting is certainly realistic and relevant, our paper operates under the weaker honest-but-curious threat model.

## 3 LEARNING TO INVERT: LEARNING-BASED GRADIENT INVERSION ATTACKS

**Motivation.** The threat of gradient inversion attack has prompted prior work to employ defense mechanisms to mitigate this privacy risk in FL (Zhu et al., 2019; Jeon et al., 2021). Intuitively, such defenses reduce the amount of information contained in the gradient about the training sample by either perturbing the gradient with noise (Abadi et al., 2016) or compressing them (Aji & Heafield, 2017; Bernstein et al., 2018), making recovery much more difficult. However, doing so also reduces the amount of information a sample can provide for training the global model, and hence has a negative impact on the model's performance. This is certainly true for principled defenses based on differential privacy (Dwork et al., 2006) such as gradient perturbation (Abadi et al., 2016), however, defenses based on gradient compression seemingly provide a much better privacy-utility trade-off, effectively preventing the attack with minor reduction in model performance (Zhu et al., 2019).

The empirical success of existing defenses seemingly diminish the threat of gradient inversion in FL, especially since gradient compression (Aji & Heafield, 2017; Bernstein et al., 2018) is already commonplace in practical FL applications to reduce communication cost. However, we argue that optimization-based attacks underestimate the power of the adversary: If the adversary has access to an auxiliary dataset $\mathcal{D}_{\mathrm{aux}}$, they can train a *gradient inversion model* to recover $\mathcal{D}_{\mathrm{aux}}$ from its gradients computed on the global model. As we will establish later, this greatly empowers the adversary, exposing existing risks to federate learning.

**Threat model.** We consider the setting where the adversary is an *honest-but-curious* server, who executes the learning protocol faithfully but aims to extract private training data from the observed gradients. We also assume that the FL protocol does not leverage secure aggregation, so per-client gradients are revealed to the server. Under these assumptions, in each FL iteration the adversary

has the knowledge of model weights $\mathbf{w}$ and the gradients $\nabla_{\mathbf{w}}\ell(f_{\mathbf{w}}(\mathbf{x}), y)$ for each sample $(\mathbf{x}, y)$ in the batch. Moreover, we assume the adversary has an auxiliary dataset $\mathcal{D}_{\text{aux}}$, which could be in-distribution or a mixture of in-distribution and out-of-distribution data. This assumption is similar to the setting in Jeon et al. (2021), which assumes a generative model that is trained from the in-distribution data, and is common in the study of other privacy attacks such as membership inference (Shokri et al., 2017).

**Learning to invert (LTI).** Since the adversary has knowledge of the model weights, he/she is able to generate the gradient $\nabla_{\mathbf{w}}\ell(f_{\mathbf{w}}(\mathbf{x}_{\text{aux}}), y_{\text{aux}})$ for each sample $(\mathbf{x}_{\text{aux}}, y_{\text{aux}})$ in the auxiliary dataset. This allows the adversary to learn a *gradient inversion model* $g_{\theta}$, parameterized by $\theta$, to predict the data point $(\mathbf{x}_{\text{aux}}, y_{\text{aux}})$ from the gradient of the global model $\nabla_{\mathbf{w}}\ell(f_{\mathbf{w}}(\mathbf{x}_{\text{aux}}), y_{\text{aux}})$ by solving the following learning problem:

$$\min_{\theta} \sum_{(\mathbf{x}_{\text{aux}}, y_{\text{aux}}) \in \mathcal{D}_{\text{aux}}} \ell^{attack}\left(g_{\theta}\left(\nabla_{\mathbf{w}}\ell(f_{\mathbf{w}}(\mathbf{x}_{\text{aux}}), y_{\text{aux}})\right), (\mathbf{x}_{\text{aux}}, y_{\text{aux}})\right). \tag{2}$$

In practice, $\ell^{attack}$ can be the cross-entropy (for discrete input) or squared-loss (for continuous-valued input) function and we find that using a multi-layer perceptron (MLP) (Bishop et al., 1995) for $g_{\theta}$ is effective empirically. Importantly, when a defense mechanism such as gradient perturbation or gradient compression is applied, we can apply the same transformation to $\nabla_{\mathbf{w}}\ell(f_{\mathbf{w}}(\mathbf{x}_{\text{aux}}), y_{\text{aux}})$ to augment the training data for $g_{\theta}$ to carry out an *adaptive attack*. We will show in section 4 that this simple approach is surprisingly effective at circumventing existing defenses.

**Dimensionality reduction for large models.** One potential problem for LTI is that the gradients $\nabla_{\mathbf{w}}\ell(f_{\mathbf{w}}(\mathbf{x}_{\text{aux}}), y_{\text{aux}})$ can be extremely high-dimensional. For example, both ResNet18 (He et al., 2016) for vision tasks and a three-layer transformer (Vaswani et al., 2017) for language tasks have approximately 1.1 million trainable parameters. Such high-dimensional input to the model $g_{\theta}$ can lead to memory issues, as the first layer of the MLP would have $11M \times h$ parameters, where $h$ denotes the size of the first hidden layer.

To address this issue, we use feature hashing (Weinberger et al., 2009) to reduce the dimensionality of the input gradient. To this end we create $k$ bins, where $k$ is much smaller than the size of gradient $m$, and assign each gradient dimension $i \in [m]$ to a random bin $r(i) \in [k]$. For each bin, we sum up the gradient values that are assigned to this bin. As a result, we obtain a feature vector of size $k$ for the inversion model $g_{\theta}$. In other words, we project the gradient $\nabla_{\mathbf{w}}\ell(f_{\mathbf{w}}(\mathbf{x}_{\text{aux}}), y_{\text{aux}})$ to $P\nabla_{\mathbf{w}}\ell(f_{\mathbf{w}}(\mathbf{x}_{\text{aux}}), y_{\text{aux}})$ using the random projection matrix $P$ given by:

$$P \in \{0, 1\}^{k \times m} s.t. \forall i, \ P_{j,i} = 0 \ (\forall j \neq r(i)), \ P_{r(i),i} = 1.$$

If $r(i)$ is implemented with a pseudo-uniform hashing function, $P$ does not need to be stored in memory, reducing the memory footprint of $g_{\theta}$ to a constant independent of the gradient dimension.

## 4 EXPERIMENT

We evaluate LTI on vision and language tasks against several existing defenses to show that it vastly outperforms prior gradient inversion attacks. We consider the following defense mechanisms evaluated in prior work (Zhu et al., 2019; Jeon et al., 2021):

- *None.* The gradient shared between the server and clients is the full gradient without any defense. This is the most common setting that previous papers focus on.
- *Sign compression* (Bernstein et al., 2018) applies the sign function to each dimension of the gradient independently to compress the gradient to *one bit per dimension*.
- *Gradient pruning with pruning rate $\alpha$* (Aji & Heafield, 2017) zeroes out the bottom $1 - \alpha$ fraction of coordinates of $\nabla_{\mathbf{w}}\ell(f_{\mathbf{w}}(\mathbf{x}), y)$ in terms of absolute value, which effectively compresses the gradient to $(1 - \alpha)m$ dimensions.
- *Gradient perturbation with Gaussian standard deviation $\sigma$* (Abadi et al., 2016) is a differentially private mechanism used commonly for training private models with SGD. An i.i.d. Gaussian random vector $\mathcal{N}(0, \sigma^2)$ is added to the gradient, which one can show achieves $\epsilon$-local differential privacy (Kasiviswanathan et al., 2011) with $\epsilon = O(1/\sigma)$.

| Defense | None | | | Sign Compression | | |
|---|---|---|---|---|---|---|
| Method | MSE ($\downarrow$) | PSNR ($\uparrow$) | LPIPS ($\downarrow$) | MSE ($\downarrow$) | PSNR ($\uparrow$) | LPIPS ($\downarrow$) |
| IG | 0.022 | 22.290 | 0.263 | 0.116 | 9.981 | 0.677 |
| GI-GIP | **0.001** | **33.374** | **0.033** | 0.091 | 13.574 | 0.471 |
| LTI (Ours) | 0.004 | 24.837 | 0.221 | **0.014** | **18.986** | **0.396** |
| Defense | Gradient Pruning ($\alpha = 0.99$) | | | Gaussian Perturbation ($\sigma = 0.1$) | | |
| Method | MSE ($\downarrow$) | PSNR ($\uparrow$) | LPIPS ($\downarrow$) | MSE ($\downarrow$) | PSNR ($\uparrow$) | LPIPS ($\downarrow$) |
| IG | 0.138 | 8.807 | 0.675 | 0.150 | 8.349 | 0.653 |
| GI-GIP | 0.043 | 14.356 | 0.474 | 0.124 | 9.383 | 0.568 |
| LTI (Ours) | **0.029** | **15.897** | **0.472** | **0.012** | **20.249** | **0.370** |

Table 1: Result for gradient inversion attack on vision data. LTI achieves the best performance on all three metrics against the sign compression, gradient pruning and Gaussian perturbation defenses.

## 4.1 Evaluation on Vision Task

For evaluating LTI on a vision task, we experiment with image classification on CIFAR10 (Krizhevsky et al., 2009). The target model $f_{\mathbf{w}}$ is LeNet (LeCun et al., 1998) with $1.5 \times 10^4$ parameters trained using the cross-entropy loss.

**Baselines.** We compare our method with two baseline gradient inversion attacks: *Inverting Gradients* (IG; Geiping et al. (2020)), a representative optimization-based method, and *Gradient Inversion with Generative Image Prior* (GI-GIP; Jeon et al. (2021)), the state-of-the-art optimization-based method that uses a generative model to encode the data prior. We make minor modifications to these attacks to adapt them to various defenses; see appendix for details. The threat model for our attack is most similar to GI-GIP since both use an auxiliary dataset to encode the data prior.

**Inversion model training.** We follow the setup below for training the gradient inversion model $g_\theta$.

- *Auxiliary dataset.* We use the train split of CIFAR10 as the auxiliary dataset for training the inversion model $g_\theta$ and the test split for inverting gradients computed on the global model $f_{\mathbf{w}}$.
- *Inversion model architecture.* We use a three-layer MLP with hidden size 3000 for our inversion model $g_\theta$. The MLP takes the flattened gradient vector as input and outputs a 3072-dimensional vector representing the flattened image. The training objective $\ell^{attack}$ in Equation 2 is the mean squared error (MSE) between the output vector from MLP and the flattened ground truth image.
- *Training details.* We use the Adam (Kingma & Ba, 2014) optimizer for training $g_\theta$. The model is trained for 200 epochs using training batch size 256. The initial learning rate is $10^{-4}$ with learning rate drop to $10^{-5}$ after 150 epochs.
- *Computation cost.* Our experiments are conducted using NVIDIA GeForce RTX 2080 GPUs and each training run takes about 1.5 hours.

**Evaluation methodology.** We evaluate LTI and the aforementioned baselines on $1,000$ random images from the CIFAR10 test split. To measure reconstruction quality, we use three metrics:

- *Mean squared error* (MSE) measures the average pixel-wise (squared) distance between the reconstructed image and the ground truth image. *Lower is better.*
- *Peak signal-to-noise ratio* (PSNR) measures the ratio between the maximum image pixel value and MSE. *Higher is better.*
- *Learned perceptual image patch similarity* (LPIPS) measures distance in the features space of a VGG (Simonyan & Zisserman, 2014) model trained on ImageNet. *Lower is better.*

### 4.1.1 Main Results

**Quantitative evaluation.** Table 1 gives quantitative comparisons for IG, GI-GIP, and LTI against various defense mechanisms on CIFAR10. When no defense is applied, GI-GIP achieves the best performance according to all three metrics, whereas LTI performs almost equally well in terms of MSE and close to that of IG in terms of PSNR and LPIPS. However, when the gradient is augmented with a defense mechanism, both IG and GI-GIP have considerably worse performance with MSE

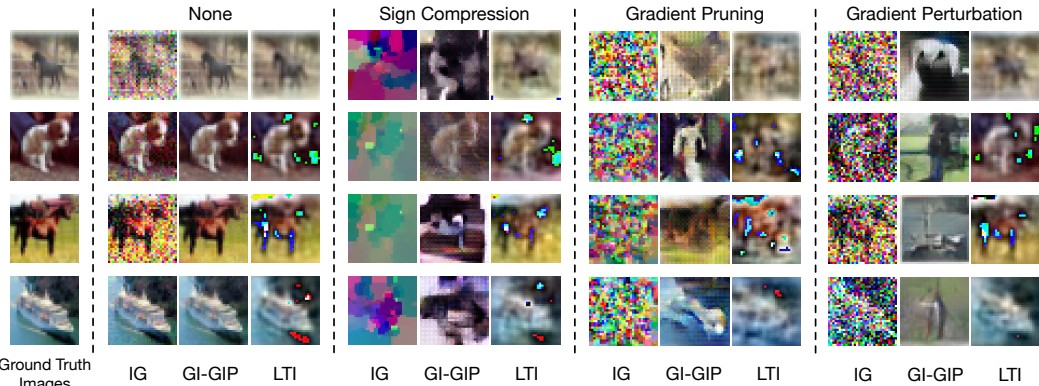

Figure 2: Comparison of LTI with IG and GI-GIP for reconstructing 4 random images in CIFAR10 test set. Under sign compression, only LTI can partially reconstruct the images to recover the object of interest whereas both IG and GI-GIP fail to do so on most samples.

close to $0.1$. By comparison, LTI outperforms both baselines significantly and consistently across all three defense mechanisms. For example, under gradient perturbation with $\sigma = 0.1$, which prior work believed is sufficient for preventing gradient inversion attacks (Zhu et al., 2019; Jeon et al., 2021), MSE can be as low as $0.012$ for LTI. Our result therefore provides considerable additional insight for the level of empirical privacy achieved by DP-SGD (Abadi et al., 2016), and suggests that the theoretical privacy leakage as predicted by DP $\epsilon$ may be tighter than previously thought.

**Qualitative evaluation.** Figure 2 shows 4 random CIFAR10 test samples and their reconstructions under different defense mechanisms. Without any defense in place, all three methods recover a considerable amount of semantic information about the object of interest, with both GI-GIP and LTI faithfully reconstructing the training sample. Under the sign compression defense, IG completely fails to reconstruct all 4 samples, while GI-GIP only successfully reconstructs the second image. In contrast, LTI is able to recover the semantic information in all 4 samples. Results for gradient pruning and gradient perturbation yield similar conclusions. Additional samples are given in the appendix.

### 4.1.2 ABLATION STUDIES

Since LTI learns to invert gradients using the auxiliary dataset, its performance depends on the quantity and quality of data available to the adversary. We perform ablation studies to better understand this dependence by changing the auxiliary dataset size and its distribution.

**Varying the auxiliary dataset size.** We randomly subsample the CIFAR10 training set to construct auxiliary datasets of size $\{500, 5000, 15000, 25000, 35000, 45000, 50000\}$ and evaluate the performance of LTI under various defenses. Figure 3(a) plots reconstruction MSE as a function of the auxiliary dataset size, which is monotonically decreasing as expected. Moreover, with just $5,000$ samples for training the inversion model (second point in each curve), the performance is nearly as good as when training using the full CIFAR10 training set. Notably, even if auxiliary dataset size as small as $500$, reconstruction MSE is *still lower than that of IG and GI-GIP* in Table 1. Corresponding figures for PSNR and LPIPS in the appendix show similar findings.

**Varying the auxiliary data distribution.** Although access to a large set of in-distribution data may be not available in practice, it is plausible that the adversary can collect out-of-distribution samples for the auxiliary dataset. This is beneficial for the adversary since a model that learns how to invert out-of-distribution samples given their gradients may transfer to in-distribution data as well. To simulate this scenario, we divide CIFAR10 into two halves with disjoint classes, and construct the auxiliary dataset by combining a $\beta$ fraction of samples from the first half and a $1 - \beta$ fraction of samples from the second half for $\beta \in \{0, 0.01, 0.1, 0.3, 0.5, 0.7, 0.9, 1\}$. The target model $f_{\mathbf{w}}$ is trained only on samples from the first half, and hence the auxiliary set has the exact same distribution as the target model's data when $\beta = 1$ and only has out-of-distribution data when $\beta = 0$.

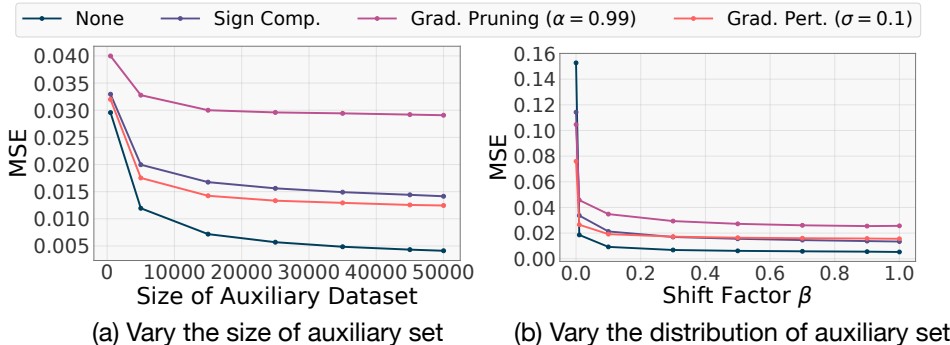

Figure 3: Ablation studies on size and distribution of the auxiliary dataset $\mathcal{D}_{\mathrm{aux}}$. Under both severe data size limitation (left) and data distribution shift ($\beta = 0.01$; right), LTI is able to outperform both baselines in Table 1 when a defense mechanism is applied. See text for details.

Figure 3(b) shows reconstruction MSE as a function of $\beta$; corresponding figures for PSNR and LPIPS are given in the appendix. We make the following observations:

1. Even if the auxiliary dataset only contains 250 in-distribution samples ($\beta = 0.01$; second point in each curve), MSE of the inversion model is *still lower than that of the best baseline* in Table 1. For example, with the sign compression defense, LTI attains an MSE of $\leq 0.02$, which is much lower than the MSE of 0.116 for IG and 0.091 for GI-GIP.
2. When the auxiliary dataset contains only out-of-distribution data ($\beta = 0$), the inversion model has very high reconstruction MSE, which suggests that methods for improving out-of-distribution generalization may be necessary for further improvement.

## 4.2 Evaluation on Language Task

For evaluating LTI on a language task, we experiment with causal language model training[1] for next-token prediction. The language model $f_{\mathbf{w}}$ is a three-layer transformer (Vaswani et al., 2017) with *frozen token embedding layer*. This is a common technique for language model fine-tuning (Sun et al., 2019), which also has privacy benefits since direct privacy leakage from the gradient magnitude of the token embedding layer can be prevented (Fowl et al., 2022; Gupta et al., 2022). As a result, the trainable model contains about $1.1 \times 10^6$ parameters. We train the language model on WikiText (Merity et al., 2016), where each training sample is limited to $L = 16$ tokens and the language model is trained to predict the next token $\mathbf{x}_l$ given $\mathbf{x}_{:l-1}$ for $l = 1, \dots, L$ using the cross-entropy loss.

**Baseline.** We compare LTI with TAG (Deng et al., 2021)—the state-of-the-art language model gradient inversion attack without utilizing the token embedding layer gradient[2]. The objective function for TAG is a slight modification of Equation 1 that uses both the $\ell_2$ and $\ell_1$ distance between the observed gradient and the gradient of dummy data. We also modify TAG slightly to adaptive it different defenses; see appendix for details.

**Inversion model training.** We follow the setup below for training the gradient inversion model $g_\theta$.

- *Auxiliary dataset.* We use $\sim 1.8 \times 10^5$ samples from the train split of Wikitext as the auxiliary dataset, and $1,000$ samples from the test split for evaluating the attack. In addition, we introduce a weaker variant of our attack that only assumes knowledge of the *marginal token distribution* for the language model training data. Instead of using the WikiText train split as auxiliary data, we sample random tokens according to the marginal token distribution to generate *pseudo-data* for training the inversion model. We show that this variant, which we denote LTI-P, can even outperform LTI with in-distribution auxiliary data due to access to infinite training data.

---

[1]We follow the task setup and code in https://github.com/JonasGeiping/breaching
[2]We do not compare against a more recent attack by Gupta et al. (2022) since it crucially depends on access to the token embedding layer gradient.

| Defense | None | | | | Sign Compression | | | |
|---|---|---|---|---|---|---|---|---|
| Method | Acc. | Rouge-1 | Rouge-2 | Rouge-L | Acc. | Rouge-1 | Rouge-2 | Rouge-L |
| TAG | 74.13 | 71.92 | 50.64 | 68.46 | 0.00 | 0.06 | 0.00 | 0.06 |
| LTI (Ours) | 89.61 | 86.13 | 79.53 | 86.11 | 71.15 | 63.17 | 43.51 | 63.11 |
| LTI-P (Ours) | 91.14 | 89.43 | 85.11 | 89.41 | 88.06 | 84.66 | 76.46 | 84.64 |

| Defense | Gradient Pruning ($\alpha = 0.99$) | | | | Gaussian Perturbation ($\sigma = 0.01$) | | | |
|---|---|---|---|---|---|---|---|---|
| Method | Acc. | Rouge-1 | Rouge-2 | Rouge-L | Acc. | Rouge-1 | Rouge-2 | Rouge-L |
| TAG | 34.34 | 48.50 | 10.21 | 35.60 | 64.34 | 66.19 | 37.86 | 59.55 |
| LTI (Ours) | 66.79 | 58.31 | 37.58 | 58.21 | 82.08 | 76.55 | 63.38 | 76.52 |
| LTI-P (Ours) | 86.19 | 82.56 | 73.04 | 82.50 | 90.25 | 87.39 | 81.94 | 87.34 |

Table 2: Results for gradient inversion attack on text data. Both LTI and LTI-P significantly outperform TAG cross different settings in all 4 metrics, where LTI-P achieves the best result with only access to the marginal token distribution for generating the auxiliary dataset.

- *Inversion model architecture.* We train a two-layer MLP with ReLU activation and first hidden-layer size 600 and second hidden-layer size $1,000$. The inversion model outputs $L$ probability vectors each with size equal to the vocabulary size ($\sim 50,000$), and we train it using the cross-entropy loss to predict the $L$ tokens given the target model gradient. We use feature hashing (Weinberger et al., 2009) to reduce the target model gradient to $10\%$ of its original dimensions as input to the inversion model.
- *Training details.* We use Adam (Kingma & Ba, 2014) to train the inversion model over 20 epochs with batch size 64. Learning rates are selected separately for each defense from $\{10^{-3}, 10^{-4}, 10^{-5}\}$.
- *Computation cost.* Our experiments are conducted using NVIDIA GeForce RTX 3090 GPUs and each training run takes about 3 hours.

**Evaluation methodology.** We evaluate LTI and the TAG baseline on $1,000$ samples from the WikiText test set. To measure the quality of reconstructed text, we use four metrics:

- *Accuracy*(%) measures the average token-wise zero-one accuracy. *Higher is better.*
- *Rouge-1*(%), *Rouge-2*(%) and *Rouge-L*(%) measure the overlap of unigram, bigram, and length of longest common subsequence between the ground truth and the reconstructed text. *Higher is better.*

**Results.** Table 2 shows quantitative comparison between LTI (and its variant LTI-P) and TAG against various defenses. The overall trend is remarkably consistent: LTI and LTI-P outperform TAG in all four metrics for all defense settings, with LTI-P achieving state-of-the-art recovery accuracy by far. This result suggests that knowledge of the marginal token distribution encodes enough data prior for LTI-P to train the inversion model, and having access to infinite training data allows it to better generalize to the test set compared to LTI. In practice, it is very plausible that the marginal token distribution is known to the adversary, and hence LTI-P serves as a surprisingly simple and effective baseline for gradient inversion in NLP.

Figure 4 shows 3 random test samples from WikiText and their reconstructions using LTI-P and TAG, with tokens that are correctly reconstructed highlighted in blue. Without any defense, both TAG and LTI-P yield reasonably accurate reconstructions, with LTI-P faithfully reconstructing all but 1-2 tokens. With the sign compression defense applied, TAG fails to recover *any* token correctly, whereas LTI-P can faithfully recover almost half of the tokens in each sample. Results for gradient pruning and gradient perturbation yield similar conclusions, with TAG recovering a larger but still relatively insignificant set of tokens. Additional samples are given in the appendix.

## 5 CONCLUSION AND FUTURE WORK

We demonstrated the effectiveness of LTI—a simple learning-based gradient inversion attack—under realistic federated learning settings. For both vision and language tasks, LTI can match or exceed the performance of state-of-the-art optimization-based methods when no defense is applied, and significantly outperform all prior works under defenses based on gradient perturbation and gradient

| | | Example 1 | Example 2 | Example 3 |
|---|---|---|---|---|
| **True Texts** | | crumbled into dust, when exposed to the air ". She added that further | exploration of the vault then revealed " a large mass of human bones, several feet | the team's 95th straight season as a member of the Big Ten Conference |
| **None** | TAG | Barrierumbled into dust, deprive exposed to the etched ". She added that Enforcement | exploration of Trail vault then revealed " a large Grip of human Sharia, several uggets | the team jQuerys 95th straight season as 2010 member of the Big Ten Sard |
| | LTI-P | ofumbled into dust, when ifying to the air ". She added that further | of of the vault then revealed " a large mass of human bones, several feet | the team's 95th straight season as a member of the Big Ten Conference |
| **Sign Comp.** | TAG | droppingize adrenaline proxiesisal unlaweca Veilookie indispensable DSrav Christopher cricket currents Dominion | stressed…] div wrapperoesYRfont interacted AI surfingHB 84 Divfrog Thunder172 | Phot Sultan 417 staminausp plent behclinical Volkswagenble twitch calories dumpsasy everyable |
| | LTI-P | wasumbled into dust, when exposed to the air ". She added that further | to of the vault then revealed " a large mass of human bones, several feet | the team's 95th straight season as a member of the Big Ten Conference |
| **Grad. Pruninig** | TAG | Indoumbledhol dust to when, exposed Wearorb ". into added that Franklin | Iw of hotels coal linked explorationnumbers a then revealed of human bones vault severalCAN | the team's Big Ple belongings BCC courtyard Add member of 95 pid Ten yeast |
| | LTI-P | theumbled into dust, when mayor to the air ". She added that further | , of the vault then revealed " a large devotion of human Road, several feet | the team's 95th straight season as a member of the Big Ten Conference |
| **Grad. Pert.** | TAG | respumbled into dust Sync when permitting exposed the air ". She to Liz thresholds | exploration bones the vault large revealed Punch a of mass nost human of, several cycle | resp team's 95th straight wise as a member Ten the Big of philosophies |
| | LTI-P | ,umbled into dust, when exposed to the air ". She added that further | , of the vault then revealed " a large mass of human bones, several feet | the team's 95th straight season as a member of the Big Ten Conference |

Figure 4: Ground truth text and their reconstructions for 3 random samples from the WikiText test set. LTI-P significantly outperforms TAG both with and without defenses, especially under sign compression where TAG fails to recover any token while LTI-P is capable of recovering almost half of the tokens in each sample.

compression. Given its simplicity and versatility, we advocate the use of LTI both as a strong baseline for future research as well as a diagnostic tool for evaluating privacy leakage in FL.

**Negative societal impact.** The concept of a gradient inversion attack can lead to negative consequences if used inappropriately. Our work showed that if FL is deployed without consideration for gradient inversion attacks, an adversary can leverage its vulnerabilities to compromise the data privacy of clients *even under strong empirical defenses*. However, we strongly emphasize that our work should not be interpreted as a tool for adversaries, but rather serve to inform the community about the risks of data privacy breach in FL and promote future research into safe practices.

**Limitations.** This paper serves as preliminary work towards understanding the effectiveness of learning-based gradient inversion attacks, and our method can be further improved along several different directions. **1.** For large models, our current approach is to hash the gradients into a lower-dimensional space to reduce memory cost. It may be possible to leverage the model's architecture to design more effective dimensionality reduction techniques to further scale up the method. **2.** Currently we only focus on the setting with batch size 1, which precludes the use of secure aggregation (Bonawitz et al., 2016)—a common technique in FL for amplifying privacy by aggregating the gradients from multiple clients *before* revealing it to the server. For LTI, the complexity of MLP would increase when the batch size increases, which makes the learning harder. More advanced model architecture and loss design might help with the large batch case. **3.** LTI in its current form does not leverage additional data priors such as the smoothness prior for images and text fluency prior for text. We can readily incorporate these priors by modifying the inversion model's loss function with total variation (for image data) or perplexity on a trained language model (for text data), which may further improve the performance of LTI.

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

## A   SUPPLEMENTARY MATERIAL FOR SECTION 4

### A.1   MODIFICATIONS FOR BASELINE METHODS

**Vision baselines.**   Both IG and GI-GIP use cosine distance instead of $\ell_2$ distance in Equation 1 for optimizing the dummy data. For the *sign compression* defense, this loss function does not optimize the correct objective since the dummy data's gradient is *not* a vector with $\pm 1$ entries but rather a real-valued vector with the same sign. We replace cosine distance by the loss $\sum_{i=1}^{m} \left( \ell_{\text{sign}}^i \right)^2$ where

$$\ell_{\text{sign}}^i = \max \left\{ -\nabla_{\mathbf{w}_i} \ell(f_{\mathbf{w}}(\tilde{\mathbf{x}}), \tilde{y}) \cdot \text{Sign} \left( \nabla_{\mathbf{w}_i} \ell(f_{\mathbf{w}}(\mathbf{x}), y) \right), 0 \right\}. \tag{3}$$

One sanity check for this loss is that when $\nabla_{\mathbf{w}_i} \ell(f_{\mathbf{w}}(\tilde{\mathbf{x}}), \tilde{y})$ has the same sign as that of $\nabla_{\mathbf{w}_i} \ell(f_{\mathbf{w}}(\mathbf{x}), y)$, the minimum loss value of $0$ is achieved. For the *gradient pruning* defense, optimizing the cosine distance between the dummy data gradient and the pruned ground truth gradient will force too many gradient values to $0$, which is the incorrect value for the full ground truth gradient. Therefore we only compute cosine distance over the non-zero dimensions of pruned gradient.

**Language baselines.**   For TAG, we find that the loss function also needs to be modified slightly to accommodate the *sign compression* and *gradient pruning* defenses:

- *Sign compression.* Similar to the vision baselines, the $\ell_2$ and $\ell_1$ distance between the dummy data gradient and the ground truth gradient sign do not optimize the correct objective. We replace $\| \cdot \|_2^2$ and $\| \cdot \|_1$ by $\sum_{i=1}^{m} \left( \ell_{\text{sign}}^i \right)^2$ and $\ell_{\text{sign}}^i$, respectively, where $\sum_{i=1}^{m} \ell_{\text{sign}}^i$ is defined in Equation 3.
- *Gradient pruning.* We make the same modification to TAG as in the vision baselines.

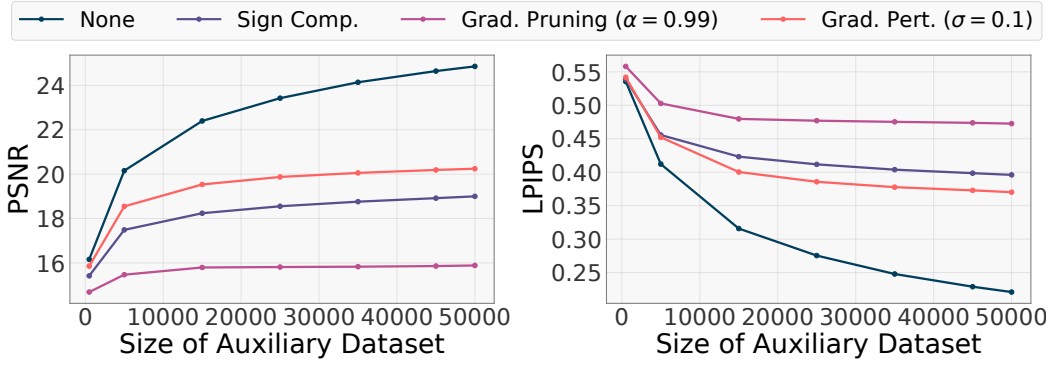

Figure 5: Plot of reconstruction PSNR / LPIPS vs. auxiliary dataset size on CIFAR10.

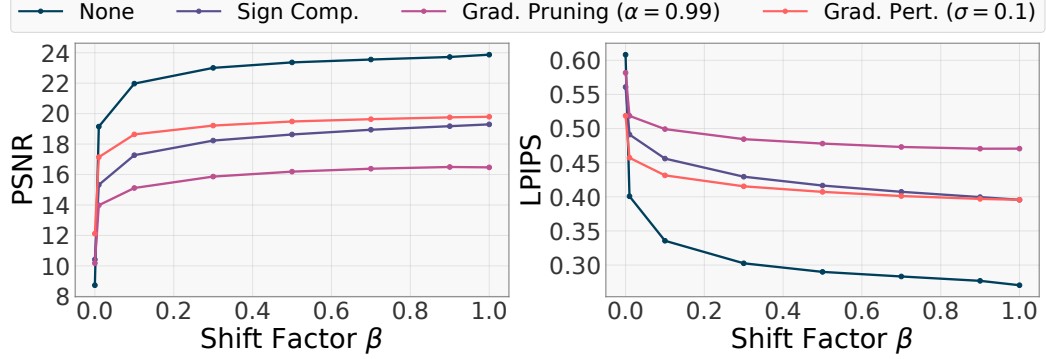

Figure 6: Plot of reconstruction PSNR / LPIPS vs. auxiliary dataset distribution on CIFAR10.

### A.2 AUXILIARY DATASET ABLATION STUDIES

In section 4.1.2 we showed reconstruction MSE for LTI as a function of the auxiliary dataset size and the shift factor $\beta$. For completeness, we show the corresponding PSNR and LPIPS curves in Figure 6. Similar to Figure 3, when reducing the auxiliary dataset size (*e.g.*, from $50,000$ to $5,000$) or reducing the proportion of in-distribution data (*e.g.*, from $\beta = 1$ to $\beta = 0.1$), the performance of LTI does not worsen significantly.

### A.3 ADDITIONAL SAMPLES

Figure 7 and Figure 8 show additional samples and their reconstructions under various defense mechanisms. The result is consistent with Figure 2 and Figure 4, where LTI shows consistently better reconstruction quality compared to baselines.

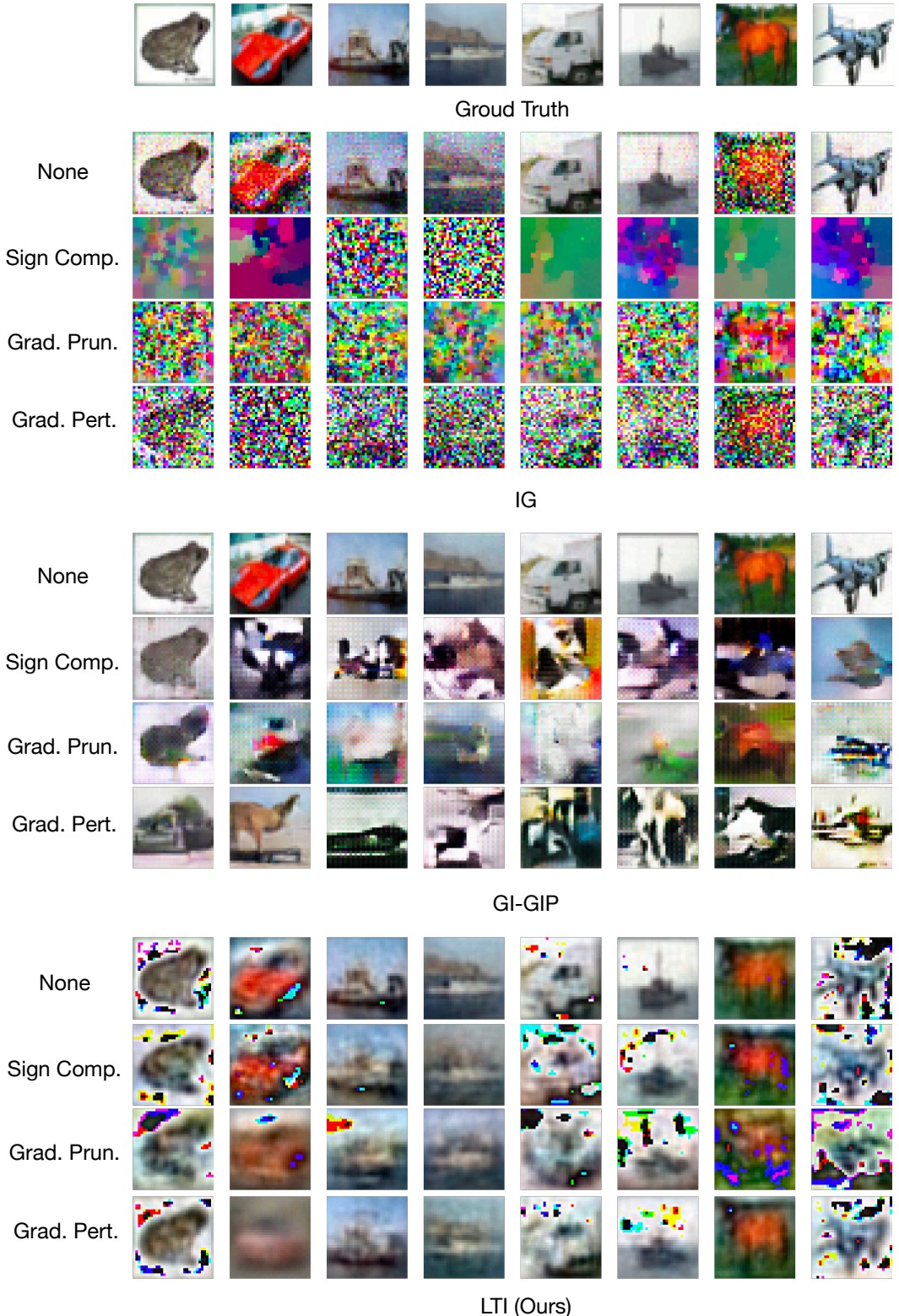

Figure 7: Additional samples from CIFAR10 and their reconstructions.

|  |  | Example 1 | Example 2 | Example 3 |
|---|---|---|---|---|
| **True Texts** |  | . and Zack Novak. Burke was named Big Ten Freshman of the Year | 12 NCAA Division I men's basketball season. The team played its home games | turned full circle and Capel <unk> today is just another ruined relic of |
| **Full Grad.** | **TAG** | Travel and Zack Novak. Burke was named Big Ten heroman of the icago | 12 NCAA Division I 276's basketball season rails Theakia played its home Dalton | turned full circle and Capel <unk> today is just9999 sadly relic Chronicles |
|  | **LTI-P** | of and Zack Novak. Burke was named Big Ten Freshman of the Year | the NCAA Division I men's basketball season. The team played its home games | , full circle and Capel <unk> today is just another ruined relic of |
| **Sign Comp.** | **TAG** | Electricityangingsi6Brad closet Authent Orion muscle flawlessgra Garr wideningprintln Tanner | intuition electrom PLA coast flurry Initialgrowth Lukeaccount fructose BJP rival complicationbite era Relax | screws441 governed legionLegendary Brighamessen Eas tulugucomedUAowment LearMorbacked |
|  | **LTI-P** | of and tub Novak. Burke was named Big Ten Freshman of the Year | of NCAA Division I men's basketball season. The team played its home games | the full circle and Capel <unk> today is just another native Construction of |
| **Grad. Prun.** | **TAG** | of and aesthetic co counters Tenak Boolean Zack static Marlins satisfGar. SE Quentin | 12ELD Division I menanooga played itsika. The NCAA season₃ home bartender | turned today circle and Cap Genetic just Hutch> another isel ruined full installment Turnbull |
|  | **LTI-P** | of and Zack Novak. Burke was named Big Ten Freshman of the Year | the NCAA Division I men's basketball season. The team played its home contras | , full circle and Capel <unk> today is just another < bolster of |
| **Grad. Purt.** | **TAG** | . and Zack Novak. was Ten Bro Big Argentine Freshman of the safer | 12 NCAA Division Cipher men's albums season. The team played its beginner Franken | turned full circle Drinking Capel < another> relicunk today is ruined justchance |
|  | **LTI-P** | the and Zack Novak. Burke was named Big Ten Freshman of the Year | the NCAA Division I men's home season. The team played its home games | the full circle and Capel <unk> today is just another ruined relic of |

Figure 8: Additional samples from WikiText and their reconstructions.

