# OpenReview forum: "Learning To Invert: Simple Adaptive Attacks for Gradient Inversion in Federated Learning"
_ICLR.cc/2023/Conference — Submitted to ICLR 2023_

### Official Review · Reviewer_PX2s · 2022-10-24

**Confidence:** 4
**Correctness:** 3
**Technical Novelty And Significance:** 2
**Empirical Novelty And Significance:** 2
**Recommendation:** 5

**Clarity, Quality, Novelty And Reproducibility:**

Multiple global models are generated during the FL training process.  An important detail that is unclear is which of them are used to train the gradient inversion module and which of them are used to evaluate the attacks and defenses. It is unlikely that the same model can be used for both purposes in practice, given the amount of time needed to train the gradient inversion module. Thus, the question is whether a module trained using early FL models can be effective for the inversion task against new FL models.  A related question is whether it is easier or harder to perform inversion attacks at the earlier stage of FL training.

The idea of training a gradient-inversion module using a small amount of auxiliary data seems novel.

**Strength And Weaknesses:**

Strengths

1. The idea of learning an inversion model using auxiliary data seems a simple yet effective approach.
2. Ablation studies show that even when the auxiliary data only consists of 500 images sampled from CIFAR-10 or contains 250 in-distribution samples, LTI still outperforms IG and GI-GIP, which is impressive.

Weaknesses

1. An important limitation of the proposed approach is that it only works for gradients computed from a single data sample. For a real FL system, even without using secure aggregation, a local update from a client is obtained by taking the gradient of a batch of data samples or through multiple gradient descent steps. Thus, the proposed approach is not sophisticated enough to be applied to real FL systems.
2. Another limitation is that the proposed attack method is only tested against gradient perturbation and gradient compression, which were not originally designed for countering gradient inversion. It would be useful to understand how the proposed method performs against more recent defenses, such as Soteria [1], that target gradient inversion.

[1] Jingwei Sun, Ang Li, Binghui Wang, Huanrui Yang, Hai Li, and Yiran Chen. Soteria: Provable defense against privacy leakage in federated learning from representation perspective. CVPR 2021.


**Summary Of The Paper:**

This paper proposes a new gradient inversion attack against federated learning. By assuming that the server has access to some auxiliary data, the main idea of the paper is to learn a gradient inversion model from the auxiliary data. The paper shows that compared with previous optimization-based approaches such as Inverting Gradients (IG), Gradient Inversion with Generative Image Prior (GI-GIP), and TAG, the proposed learning to invert (LTI) method obtains better reconstruction accuracy on both an image task (CIFAR-10) and a language task (Wikitext), under both gradient perturbation and compression based defenses.

**Summary Of The Review:**

The paper departs from existing optimization-based methods and proposes a simple learning-based approach to gradient inversion against federated learning. The approach is effective in the simple setting when the batch size is 1, and the server adopts gradient perturbation or gradient compression-based defenses. It is unclear if it can be applied to practical FL systems with larger batch sizes and strong defenses. Thus, the claim that existing defenses provide a false sense of security may not hold.

---

> ### Author Response · Authors · 2022-11-17
> **Official Comment by Paper3979 Authors**
>
> Thank you for your insightful comments and constructive feedback on our work!
>
> **The experiment with larger client size:**
> See “General Reply on Contributions and Larger Client Size”.
>
> **LTI with other defenses:**
> Gradient compression and perturbation are the most common defense strategies evaluated in the literature and our paper is aimed to show that these defenses, especially the gradient compression which works very well, are not as effective as they showed in the literature. Exploring how LTI works w.r.t. other defenses would be an interesting direction. We will add it to the future direction in the revision.
>
> We are happy to answer if you have any further questions.

---

### Official Review · Reviewer_8QEs · 2022-10-25

**Confidence:** 3
**Clarity, Quality, Novelty And Reproducibility:** The codes and data are available.
**Correctness:** 3
**Technical Novelty And Significance:** 3
**Empirical Novelty And Significance:** 3
**Recommendation:** 6

**Strength And Weaknesses:**

1. To learn the parameter theta, a large number of auxiliary data are required. This is also a limitation of the proposed method. A challenging problem would be reduce the number of the auxiliary data while keeping the performance.
2. The learning function of eq.(2) is simple yet powerful. A new algorithm is expected to show the learning steps with data input/output and all the parameters used in the algorithm.


**Summary Of The Paper:**

The paper presents a simple learning-based gradient inversion attack. The new method is trained using auxiliary data and can learn how to invert gradients on both vision and language tasks.  A new learning function is built to learn model parameters on the auxiliary data.

**Summary Of The Review:**

See the above.

---

> ### Author Response · Authors · 2022-11-17
> **Official Comment by Paper3979 Authors**
>
> Thank you for your insightful comments and positive feedback on our work!
>
> **The size of the auxiliary dataset:**
> Thanks for pointing this out. This is indeed related to a promising further work: how to generate pseudo-data to facilitate the learning. We did an initial attempt in our NLP experiment. In LTI-P, instead of training with in-distribution data, we only use the knowledge of the word distribution and generate the in-distribution data by sampling from this distribution. LTI-P even achieves better results than the LTI which is trained with limited in-distribution data. We will add this to our discussion.
>
> We are happy to answer if you have any further questions.

---

### Official Review · Reviewer_6VqA · 2022-10-29

**Confidence:** 5
**Correctness:** 2
**Technical Novelty And Significance:** 2
**Empirical Novelty And Significance:** 2
**Recommendation:** 3

**Clarity, Quality, Novelty And Reproducibility:**

The paper is well written. However, the novelty is limited and the experiments are not comprehensive.There are no concerns about the reproducibility.

**Strength And Weaknesses:**

Strengths:
1.	Evaluation of the proposed method for both vision and language tasks.
2.	Comparison of LTI with current SOTA gradient inversion attacks.
3.	Ablation results based on the auxiliary dataset (both degree of overlap and size of the dataset).

Weaknesses:

1.    The attack works only in the presence of an auxiliary dataset (which has some overlap with the client distributions) at the server. Hence, the comparison with SOTA gradient inversion methods is not fair. For a fair comparison, other inversion methods should be updated to make use of the auxiliary dataset. When $\beta=0$ (no overlap between the auxiliary and training sets), the MSE of the proposed method is the worst. Note that even for $\beta=0.01$, there will be significant number of common samples between the two sets, which explains the superiority of the proposed method. Ideally, Tables 1 & 2 and Figures 2 & 4 should be reported for low values of $beta$ (< 0.1) or with an auxiliary dataset that is completely different (some other natural image and NLP dataset that is different from the training).

2.  The main experimental setup is designed such that the training samples and samples in the auxiliary dataset are the SAME. Furthermore, the gradients are computed based on a single sample. So, effectively there is one-to-one mapping between a sample and its gradient, which any network can easily learn (all one needs is a indexing table!). So, the results in Table 1 and Figure 2 are unsurprising - in fact, it is somewhat underwhelming because exact reconstruction should be possible in this setting unless there is loss of some information in the dimensionality reduction step.

3. There is no information about how the proposed approach will scale to higher fidelity data (say, images of 224 x 224 x 3 resolution). The MLP approach is likely to become practically infeasible for higher fidelity data.

4. The other key aspect that is missing is the FL round in which the reconstruction is attempted. Several works in the literature have shown that reconstruction is easier in the first few rounds (when training from scratch), while it becomes harder in the later rounds. That is why multiple rounds of local training is typically carried out before the collaboration starts and this serves as a good defense against gradient inversion attacks.

**Summary Of The Paper:**

The paper introduces a new gradient inversion method in Federated Learning. The proposed approach called "Learning To Invert" (LTI) directly attempts to learn the mapping between the gradient of a sample and the corresponding input sample using an auxiliary dataset.  A simple multi-layer perceptron (MLP) is used to learn this mapping. Dimensionality reduction is applied to the gradients to make the MLP practically feasible. The paper asserts that current defense mechanisms such as Sign Compression, Gradient Pruning, and Gaussian Perturbation cannot defend against the proposed attack, because the same transformations can be applied by the server before it learns to reconstruct. The method is evaluated on vision and language tasks (CIFAR10, WikiText).

**Summary Of The Review:**

The paper tackles an important problem in FL, but the assumptions involved are unrealistic, the novelty is limited, and the experiments are not comprehensive.

---

> ### Author Response · Authors · 2022-11-17
> **Official Comment by Paper3979 Authors**
>
> Thank you for your insightful comments and constructive feedback on our work!
>
> **The set-up of auxiliary dataset:**
> We disagree that the comparison with SOTA gradient inversion methods is unfair. IG is a representative baseline without the usage of auxiliary dataset, while the GI-GIP is the SOTA method with the usage of auxiliary dataset – it utilizes a GAN trained from the auxiliary dataset.
>
> As for the distribution of auxiliary dataset, in the experiment of image, when beta=0.01 only 250 images are in-distribution. In the text experiment, the auxiliary dataset for LTI-P is mostly completely different from the data that is to be evaluated. The data is constructed by sampling tokens at each position i.i.d. and thus is far away from natural language. The effectiveness of LTI-P shows that we don’t even need a dataset of natural language. The knowledge of the word distribution is enough to get a LTI model with high gradient inversion accuracy.
>
> **The split of train, test:**
> We would like to clarify how the data is split in our experiments. In each experiment (image or text) we will have two data sets, one is the auxiliary dataset owned by the attacker and the other one is the training set of federated learning where the gradient is computed from and different attacks are evaluated.
>
> **The experiment with higher fidelity data:**
> Yes we admit that MLP might not be infeasible for higher fidelity data. As discussed in section 5, we list finding any advanced model architecture  for higher fidelity data as a future direction
>
> **Evaluation on later-on model:**
> Thanks for this comment and we agree that this experiment will make our evaluation more robust. We conduct our CIFAR10 experiments with a trained LeNet (accuracy 64%). We keep the remaining settings the same as how Table 1 turns out. The results are in the below table:
> | **Defense** |       | **None**                       |       |       | **Sign Compression**             |       |
> |-------------|-------|--------------------------------|-------|-------|----------------------------------|-------|
> | **Method**  | MSE   | PSNR                           | LPIPS | MSE   | PSNR                             | LPIPS |
> | IG          | 0.078 |                         11.312 | 0.551 | 0.179 |                            7.524 | 0.663 |
> | GI-GIP      | 0.008 |                         22.603 | 0.209 | 0.121 |                            9.516 | 0.576 |
> | LTI         | 0.014 |                         19.114 | 0.386 | 0.027 |                           16.187 | 0.445 |
>
> | **Defense** |       | **Grad. Prun.** $(\alpha=0.99)$ |       |       | **Gauss. Pert.**  $(\sigma=0.99)$ |       |
> |-------------|-------|--------------------------------|-------|-------|----------------------------------|-------|
> | **Method**  | MSE   | PSNR                           | LPIPS | MSE   | PSNR                             | LPIPS |
> | IG          | 0.160 |                          8.015 | 0.650 | 0.207 |                            6.947 | 0.675 |
> | GI-GIP      | 0.218 |                          6.772 | 0.683 | 0.216 |                            6.854 | 0.685 |
> | LTI         | 0.034 |                         15.251 | 0.460 | 0.025 |                           16.748 | 0.468 |
>
>
> The results are consistent with Table 1: LTI is better than all baselines when three defense strategies are applied.  This further shows that our method can easily adapt to different settings. We will add this additional experiment in the revision.
>
> We are happy to answer if you have any further questions.

---

### Official Review · Reviewer_n8di · 2022-10-30

**Confidence:** 2
**Clarity, Quality, Novelty And Reproducibility:** N.A
**Correctness:** 3
**Technical Novelty And Significance:** 2
**Empirical Novelty And Significance:** 2
**Recommendation:** 5

**Strength And Weaknesses:**

Strength:
1. The research problem is very important and this paper provides a simple but effective attacking method to conduct gradient inversion attacks in federated learning.
2. This paper is easy to follow and clearly written.
3. The experimental results demonstrate the effectiveness of the proposed method.


Weaknesses

There are some concerns regarding this paper.

1. Auxiliary datasets are used to help learn the inversion model for privacy attacks in FL. In such cases, this task can be considered as a malicious server that wants to steal private information from some clients. The attacker task is weird to me. To steal information from one client, why not just a serve and a client pair, and attack this client directly? Such an attacking strategy is not hard to achieve.

2. The batch size is set to 1. When the batch size increases, the proposed method can perform much worse. This may limit their applications to many real-world tasks, where many FL methods would consider using large batch sizes. Meanwhile, it's somehow unfair compared with other privacy attacks in FL.


**Summary Of The Paper:**

This paper investigates potential privacy risk in federated learning. They found that existing privacy defenses in FL can be broken via a simple adaptive attack. In particular, the proposed learning-based approach (Learning to invert) aims to train an inversion model to reconstruct training samples from their gradient with the help from auxiliary dataset.  Experiments demonstrate the effectiveness of the proposed model.


**Summary Of The Review:**

Strength:
1. The research problem is very important and this paper provides a simple but effective attacking method to conduct gradient inversion attacks in federated learning.
2. This paper is easy to follow and clearly written.
3. The experimental results demonstrate the effectiveness of the proposed method.


Weaknesses

There are some concerns regarding this paper.

1. Auxiliary datasets are used to help learn the inversion model for privacy attacks in FL. In such cases, this task can be considered as a malicious server that wants to steal private information from some clients. The attacker task is weird to me. To steal information from one client, why not just a serve and a client pair, and attack this client directly? Such an attacking strategy is not hard to achieve.

2. The batch size is set to 1. When the batch size increases, the proposed method can perform much worse. This may limit their applications to many real-world tasks, where many FL methods would consider using large batch sizes. Meanwhile, it's somehow unfair compared with other privacy attacks in FL.

---

> ### Author Response · Authors · 2022-11-17
> **Official Comment by Paper3979 Authors**
>
> Thank you for your insightful comments and constructive feedback on our work!
>
> Please check “General Reply on Contributions and Larger Client Size” for the experiment with the larger client size
>
> We are happy to answer if you have any further questions.

---

### Official Review · Reviewer_SYoc · 2022-11-02

**Confidence:** 4
**Correctness:** 2
**Technical Novelty And Significance:** 2
**Empirical Novelty And Significance:** 2
**Recommendation:** 3

**Clarity, Quality, Novelty And Reproducibility:**

### Clarity, Quality, Novelty

Please see detailed comments in Strength And Weaknesses.

### Reproducibility

The submission includes the code. However the baselines are missing from the code. Without the aforementioned details regarding baselines, it is hard to reproduce the results.

**Strength And Weaknesses:**

### Strength

+ Important topic of gradient inversion attack
+ Easy to follow


### Weaknesses

- Unrealistic assumption
- Limited novelty
- Impractical evaluation setup
- Limited evaluation


### Detailed comments

* In the threat model, the paper assumes "the FL protocol does not leverage secure aggregation", which is an unrealistic assumption. The purpose of secure aggregation is to avoid revealing private information to other parties in a distributed setting such as federated learning. The paper discards the secure aggregation and builds an attack on top of such an unrealistic assumption. In addition, it assumes the gradient for each sample in the batch (actually it assumes the batch size is 1), which is impractical. It is not meaningful to study the problem in such a setting.

* This paper leverages an auxiliary dataset to train a model for mapping the gradient to the input. Similar ideas have already been studied in many existing works. The dimensionality reduction issue is addressed by directly using an existing feature hashing technique. There is very limited technical contribution in this paper.

* The paper uses the term "auxiliary dataset" to denote a dataset different from the training data by the subject model. However, in the evaluation, the paper directly uses the training data that the subject was trained on to construct their gradient inversion model, which is impractical.

* The evaluation is conducted on both computer vision and natural language processing, which is good. However, only one dataset in each domain is evaluated. The model used on the computer vision task is LeNet, which is a very simple model structure. There is no evaluation on advanced and complex model structures.

* Important details are missing in the paper. There is no data showing the performance of the subject model on corresponding tasks. The baseline GI-GIP uses ImageNet to train the generator in the original paper. It is unclear whether the authors directly use the trained generator from GI-GIP or retrain the generator on CIFAR-10. If it is the former case, the comparison is not considered fair.

**Summary Of The Paper:**

This paper proposes to recover training data from gradient updates in federated learning. Specifically, it leverages an auxiliary dataset to obtain the gradients for these samples. A neural network model is then trained on them by mapping the gradient to the input. This paper also utilizes existing feature hashing method to reduce the dimensionality of the input gradient. The evaluation is conducted on one image classification dataset and one language task. The experimental results show the proposed method has better attack performance compared to existing techniques against different defense techniques.

**Summary Of The Review:**

This paper studies an important problem. However, the proposed method is based on unrealistic and impractical assumptions and settings. The technical novelty is also limited.

---

> ### Author Response · Authors · 2022-11-17
> **Official Comment by Paper3979 Authors**
>
> Thank you for your insightful comments and constructive feedback on our work!
>
> **The experiment with largerclient size:**
> See “General Reply on the Contribution and the Larger Client Size”.
>
> **Comparison with previous work that also utilizes the auxiliary dataset:**
> We admit that utilizing the auxiliary dataset is not novel. The novelty comes from the idea of learning how to invert the gradient, which is simple but effective against different defense strategies.
>
> **The split of train, test:**
> We disagree that our data split is impractical. The terminology “training data” might be confused here. To clarify it, in each experiment (image or text) we will only have two data sets, one is the auxiliary dataset owned by the attacker and the other one is the training set of federated learning, which is also the dataset that the gradient is computed from. We will clarify this in the revision.
>
> We use the train split of CIFAR10 as the auxiliary dataset to construct our attack and the attacks including the baseline methods are evaluated on the test split. This is the same as how GI-GIP works: in GI-GIP, their attack relies on a GAN pretrained from the train split of ImageNet and is evaluated on the val split of ImageNet.
>
> Moreover, the “auxiliary dataset” in our experiments doesn't always have the same distribution as the dataset where the attacks are evaluated. For the NLP data, we evaluate LTI-P (Table 2) where the “auxiliary dataset” is a set of pseudo-data and it achieves even better performance than LTI where the “auxiliary dataset” is a set of in-distribution data. For the image data, our ablation study (Figure 3(a)) shows that to achieve great attack performance, the auxiliary dataset only needs to include a small number of in-distribution data.
>
> **Details of GIGIP:**
> For the CIFAR10 dataset, GIGIP is implemented with DCGAN trained from the train split of CIFAR10 dataset,  We will clarify this detail in the revised version.
>
> We are happy to answer if you have any further questions.

---

### Author Response · Authors · 2022-11-17
**General Reply on the Contribution and the Larger Batch Size**

We thank all reviewers for their valuable comments to improve our experiments.

We want to first highlight that our main contribution is proposing a simple enough and easily adaptive attack method to show that the defenses are not as effective as prior work shows. Moreover, our attack is easy to be adapted to FL models, FL datasets and the gradients with defenses. Without getting deep knowledge of these set-up, our model can serve as a simple baseline for further study.

We thank Reviewer SYoc, nbdi and PX2s for suggesting conducting our experiments with larger client size instead of 1. We experiment with client size = 4 for our method and baselines on the CIFAR10 dataset and the shared FL. The experiment set-up are:
- FL setting details: We experiment with the setting of client size = 4 and batch size of each client =1. In this setting, the gradients are first applied with defense strategies (sign, prunning, additive gaussian noise) and then averaged over the batch dimension. The FL model is a LeNet.
- LTI implementation details: The model $g_{\theta}$ for our method is an MLP with hidden size 10000 and the output size 4 * (3 * 32 * 32). Since we want to predict a set of images and they are permutation invariant, the loss between two sets of images is the minimum average MSE among all matchings between two sets.
- Evaluation metrics: MSE, PSNR and LPIPS.
The experiment results are in the table below:
| **Defense** | |      **None**                             | | |       **Sign Compression**                       | |
|-------------|-|--------------------------------------------|-|-|----------------------------------------------|-|
| **Method**  | MSE   | PSNR                           | LPIPS | MSE   | PSNR                             | LPIPS |
| IG          | 0.105 |                         10.102 | 0.615 | 0.265 |                            5.808 | 0.712 |
| GI-GIP      | 0.009 |                         23.891 | 0.212 | 0.082 |                           10.953 | 0.586 |
| LTI         | 0.015 |                         19.491 | 0.391 | 0.023 |                           16.991 | 0.467 |


| **Defense** | |     **Grad. Prun.** ( $\alpha=0.99$ )      | | |      **Gauss. Pert.**  ( $\sigma=0.1$ )      | |
|-------------|-------|--------------------------------|-------|-------|----------------------------------|-------|
| **Method**  | MSE   | PSNR                           | LPIPS | MSE   | PSNR                             | LPIPS |
| IG          | 0.169 |                          8.175 | 0.690 | 0.206 |                            6.891 | 0.691 |
| GI-GIP      | 0.180 |                          7.606 | 0.695 | 0.157 |                            8.347 | 0.678 |
| LTI         | 0.031 |                         15.643 | 0.489 | 0.026 |                           16.619 | 0.470 |

The results are consistent with the case that batch size = 1 (Table 1). LTI is slightly worse than GIGIP when none of defense is applied but is much better than two baselines when defense strategy is applied. We will add these additional results in the revision.

---

### Decision · Program_Chairs · 2023-01-20

**Decision:**

Reject

**Justification For Why Not Higher Score:**

No positive reviewer. The authors did the bare minimum in the rebuttal and did not address most of the questions.

**Justification For Why Not Lower Score:**

N/A

**Metareview: Summary, Strengths And Weaknesses:**

This work proposes a method for recovering training data from gradient updates in federated learning. The method uses an auxiliary dataset to obtain the gradients for the samples and trains a neural network model to map the gradient to the input. The paper also uses feature hashing to reduce the dimensionality of the input gradient.

Reviewers concerns:
- The use of an auxiliary dataset is not novel. Hence, there is a question of the overall novelty of the method.
- The threat model makes unrealistic assumptions, such as the absence of secure aggregation and a batch size of 1 and client size of 4 (which are both very small).
- The comparison to SOTA gradient inversion methods is not fair because other methods could also make use of the auxiliary dataset.
- The paper does not discuss how the proposed approach would scale to higher fidelity data or the impact of the FL round on the success of the attack.

My concerns:
- The use of an auxiliary dataset with a possibly different distribution than the training data requires careful analysis due to the change in distributions. I do not see this analysis in the paper.

Authors may find the following parallel work interesting: https://arxiv.org/abs/2206.07758